# Multimodal Fusion Analysis of [18F]Florbetapir PET and Multiscale Functional Network Connectivity in Alzheimer's Disease

Bikesh Bimali*[†], Nigar Khasayeva*[†], Adithya Ram Ballem*[‡], Geethanjali Nagaboina*[†], Jiayu Chen*[†], Alex Fedorov[§], James J Lah[¶], Allan Levey[¶], Vince D. Calhoun*[†‡], Armin Iraji*[†]

*Tri-Institutional Center for Translational Research in Neuroimaging and Data Science (Georgia State University, Georgia Institute of Technology, Emory University), Atlanta, USA
[†]Department of Computer Science, Georgia State University, Atlanta, GA, USA
[‡]School of Electrical and Computer Engineering, Georgia Institute of Technology, Atlanta, GA, USA
[§]Center for Data Science, Nell Hodgson Woodruff School of Nursing, Emory University, Atlanta, GA, USA
[¶]Department of Neurology, Emory University School of Medicine, Atlanta, GA, USA

*Abstract*—Accumulation of amyloid-beta plaques and disruption of intrinsic brain networks are two important characteristics of Alzheimer's disease (AD), yet the relationship between amyloid accumulation and network dysfunction remains unclear. In this study, we integrated [18F]Florbetapir PET and resting-state fMRI (rsfMRI) derived Functional Network Connectivity (FNC) from 552 temporally matched longitudinal PET–rsfMRI sessions across 395 participants spanning Cognitively Normal (CN), Mild Cognitive Impairment (MCI), and AD stages. With a model order of 11, joint Independent Component Analysis (jICA) was applied to the fused PET–FNC data, identifying 11 stable components, of which 9 PET-derived components corresponded to previously characterized brain regions or networks. The multimodal analysis revealed disease progression markers, including (1) a pattern of reduced subject loadings across clinical stages (CN > MCI > AD) in white matter and cerebellar regions, reflecting structural degeneration; (2) increased amyloid accumulation in affected individuals in grey matter regions, particularly in frontal, sensorimotor, extended hippocampal, and default mode network (DMN) regions, accompanied by functional connectivity alterations that reflected both compensatory and disruptive network dynamics. We identified PET-derived components that captured distinct stages of disease progression, with the DMN component emerging as a late-stage biomarker and a white matter component showing early-stage changes with limited progression thereafter. Additionally, several components showed significant variation in loadings between APOE $\varepsilon$4 carriers and non-carriers, linking the multimodal signatures to a well-established genetic risk factor for AD.

*Index Terms*—Multimodal fusion, Alzheimer's disease, Independent Component Analysis, Amyloid-beta, Positron Emission Tomography, Resting-state fMRI, Functional Network Connectivity, APOE genotype

## I. INTRODUCTION

Alzheimer's disease (AD) is a progressive neurodegenerative disorder and a leading cause of Dementia globally. Pathologically, AD is characterized by extracellular amyloid-beta (Aβ) plaques and intracellular neurofibrillary tau tangles, which disrupt synaptic function and neural connectivity, resulting in cognitive decline, memory impairment, and reduced functional capacity [1].

Positron Emission Tomography (PET) imaging with amyloid-specific radiotracers, such as [18F]Florbetapir (FBP), has emerged as an essential tool for detecting Aβ deposition. FBP selectively binds to amyloid plaques, allowing precise quantification of amyloid burden in cortical regions and hippocampus [2]. Recent studies utilizing FBP-PET have demonstrated significant amyloid accumulation within critical brain networks such as the default mode network (DMN), salience network, left and right temporal, and frontal region, underscoring their importance in AD pathology [3]. Concurrently, resting-state functional magnetic resonance imaging (rsfMRI) has independently identified functional network connectivity (FNC) alterations in AD, particularly highlighting disrupted connectivity within the DMN, salience, and frontal networks [4], [5]; regions that also exhibit early amyloid accumulation in PET studies. These converging findings support the hypothesis that amyloid pathology and network-level dysfunction in AD are spatially and mechanistically related.

Despite considerable advancements in unimodal biomarker research, the multimodal relationship between amyloid pathology and functional network disruptions remains insufficiently explored. Amyloid-PET enables quantification of regional Aβ deposition, while resting-state FNC derived from fMRI reflects intrinsic communication between distributed brain networks. Integrating these modalities may provide critical insights into how amyloid pathology accompanies functional alterations. To date, most analyses have relied on unimodal approaches, often correlating PET-derived amyloid burden and fMRI-based FNC metrics in a post hoc manner to infer cross-modal associations [6], [7].

Even among studies that incorporate both modalities, the

A. Iraji was supported by NIH grant R01MH136665. A. Fedorov was supported by the Nell Hodgson Woodruff School of Nursing at Emory University, philanthropic funds donated to the Goizueta ADRC, and the NIH National Institute on Aging (P30AG066511).

majority of studies adopt region of interest (ROI) based analyses or focus on classification tasks, using features extracted independently from each imaging modality. Multivariate methods such as canonical correlation analysis (CCA), along with more recent machine learning and deep learning approaches, have been applied to explore relationships between modalities [8], [9], [10]. However, these methods typically treat PET and fMRI as separate data streams rather than integrating them at the feature level. Consequently, they prioritize predictive performance over interpretability and often overlook the identification of shared, spatially distributed patterns that jointly reflect molecular and functional alterations. To the best of our knowledge, no prior study has jointly analyzed FBP-PET and FNC to investigate whole-brain multimodal associations in AD.

To address this gap, we employ a multimodal fusion framework that combines FBP-PET and resting-state FNC data to capture interdependent pathological and functional processes in the brain. Specifically, we apply joint ICA to the fused PET–FNC dataset, enabling the decomposition of high-dimensional, cross-modal data into statistically independent components (ICs). A key advantage of ICA-based methods in multimodal fusion is their capacity to decompose data into underlying source signals without prior knowledge or explicit modeling assumptions, potentially revealing novel and clinically relevant biomarkers. We hypothesize that these multimodal ICs capture covariations between amyloid deposition and connectivity disruptions across subjects and sessions, indicating shared underlying biology of integrated functional and metabolic brain states or dysfunctions. By jointly analyzing Aβ pathology and functional connectivity, we aim to achieve a more comprehensive understanding of AD pathophysiology, potentially offering biomarkers reflective of both amyloid-driven pathology and associated network adaptations.

## II. MATERIALS AND METHODS

### A. Participants and Data Acquisition

Longitudinal data were obtained from the Alzheimer's Disease Neuroimaging Initiative (ADNI) database (https://adni.loni.usc.edu/), comprising 1,832 PET sessions (920 unique participants) and 2,462 resting-state fMRI sessions (1,038 unique participants). PET and fMRI sessions were temporally matched for each participant, retaining pairs with scan dates within ±180 days. After filtering, 552 matched sessions from

#### TABLE I
DEMOGRAPHICS OF THE STUDY GROUP

| Diagnosis | N (N*) | Sex | N (N*) | Age (mean ± SD) |
|---|---|---|---|---|
| CN | 292 (195) | Male | 122 (81) | 74.89 ± 7.63 |
| | | Female | 170 (114) | 71.66 ± 7.14 |
| MCI | 150 (117) | Male | 82 (65) | 75.69 ± 7.84 |
| | | Female | 68 (52) | 75.47 ± 6.19 |
| AD | 110 (83) | Male | 46 (41) | 77.70 ± 8.56 |
| | | Female | 64 (42) | 75.17 ± 7.89 |

N: Number of total sessions, N*: Number of participants

395 unique subjects were included in the analysis. Participants were divided into three diagnostic groups: Cognitively Normal (CN; 292 sessions from 195 subjects), Mild Cognitive Impairment (MCI; 150 sessions from 117 subjects), and Alzheimer's disease (AD; 110 sessions from 83 subjects). Demographics, including age and sex are summarized in Table I.

### B. PET and FNC Data Acquisition

ADNI PET images were acquired following intravenous administration of the FBP radiotracer for quantifying amyloid deposition. PET data preprocessing included spatial normalization to a standardized anatomical template using SPM (https://www.fil.ion.ucl.ac.uk/spm/), smoothing with a 10-mm Gaussian kernel, and spatial downsampling to a 3×3×3 mm³ resolution using AFNI (https://afni.nimh.nih.gov/). The PET data were downsampled to reduce the dimensionality of PET features and better align with the lower-dimensional FNC data for balanced multimodal fusion.

Similarly, rsfMRI data from ADNI underwent standard preprocessing steps, including motion and distortion correction (using FMRIB Software Library; https://fsl.fmrib.ox.ac.uk/fsl/fslwiki), slice timing correction using SPM, spatial normalization to MNI space, and spatial smoothing using a Gaussian kernel with a full-width half maximum of 6 mm, as implemented by [11]. Functional Network Connectivity (FNC) matrices were then computed using the GIFT toolbox (https://trendscenter.org/software/gift/) [12] based on the Neuromark 2.2 template [13], which defines 105 multiscale intrinsic connectivity networks (ICNs). These ICNs are grouped into seven canonical functional domains: CB (Cerebellar), VI (Visual), PL (Paralimbic), SC (Subcortical), SM (Sensorimotor), HC (Higher Cognition), and TN (Triple Network). The resulting FNC matrix for each session was a 105 × 105 correlation matrix, yielding 5,460 unique connectivity edges. Detailed information on the multiscale ICNs and subdomain definitions is available in [13], [14].

### C. Multimodal Feature Construction

To ensure balanced contributions from both modalities, PET voxels and FNC edges were standardized using z-score normalization separately. For each matched session, the PET image (voxel-wise) and FNC matrix (edge-wise) were flattened into 1D vectors and concatenated, forming a multimodal feature vector (PET + FNC). The final data matrix dimensions were 552 sessions × 73,695 multimodal features.

### D. Multimodal Independent Component Analysis (ICA)

The concatenated multimodal PET–FNC dataset was decomposed using joint ICA with the Infomax algorithm via the FIT toolbox (https://trendscenter.org/software/fit/). Model order selection for ICA was guided by the Akaike Information Criterion (AIC) and Bayesian Information Criterion (BIC) [15], both supporting 11 independent components (ICs) as optimal.

To ensure stability, ICASSO algorithm was applied with 100 bootstrap iterations, resulting in highly stable components

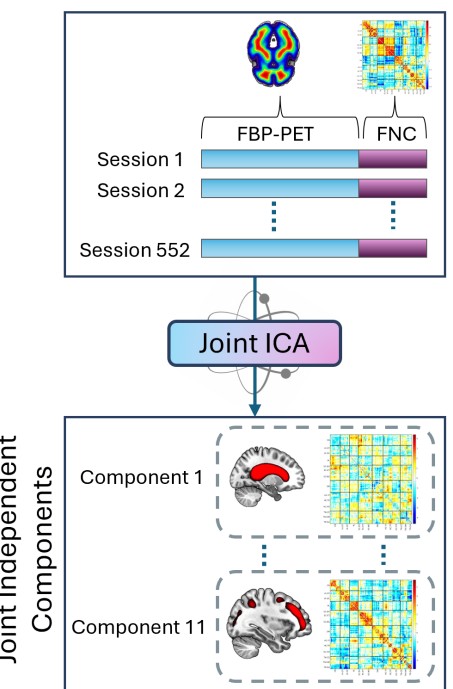

Fig. 1. Schematic of joint ICA applied to fused multimodal (PET + FNC) data, decomposing into joint independent components.

(cluster quality index $> 0.96$). Mathematically, ICA decomposes the data matrix $X$ as:

$$\underbrace{X_{n \times f}}_{\text{session} \times \text{features}} = \underbrace{A_{n \times k}}_{\text{session} \times \text{components}} \cdot \underbrace{S_{k \times f}}_{\text{components} \times \text{features}} \tag{1}$$

Where, matrix $\mathbf{A}$ represents session-specific loadings (mixing matrix), and matrix $\mathbf{S}$ contains the independent components (ICs), each with two subcomponents: (1) a PET spatial map of regional amyloid deposition and (2) an FNC matrix of functional connectivity patterns. After decomposition, PET subcomponents were remapped to identify Aβ-related signal sources, and FNC subcomponents were reshaped into $105 \times 105$ matrices (Fig. 1).

*E. Linear Mixed-Effects Model (LMM) Analysis*

To assess diagnostic group differences in IC weights while accounting for repeated measures, a linear mixed-effects model (LMM) was applied to session-specific ICA weights. To minimize site-related bias, imaging sites contributing fewer than 10 different sessions or containing sessions from only a single diagnostic group were excluded, leaving 347 sessions (241 unique individuals) for LMM analysis. The model included fixed effects for diagnosis (CN/MCI/AD), age, sex, site, race, and head motion, with a subject-specific random intercept to account for within-subject variability. The model was specified as:

$$\begin{aligned} \text{session\_loading}_{ij} \sim \ &1 + \text{dx}_i + \text{age}_i + \text{gender}_i + \text{site}_i \\ &+ \text{race}_i + \text{head\_motion}_i + (1 \mid \text{rid}_i) \end{aligned} \tag{2}$$

Here, session_loading$_{ij}$ represents the loading of the $j^{\text{th}}$ IC for the $i^{\text{th}}$ session. The variable RID denotes the unique subject identifier, allowing the grouping of multiple sessions per subject and modeling subject-specific random intercepts to capture baseline inter-subject variability.

*F. APOE ε4 Carrier-Based Group Comparison*

To examine the association between APOE $\varepsilon$4 status and multimodal component loadings, a separate group comparison was conducted using genotype data from the ADNI database, without inclusion of additional covariates. This analysis aimed to assess whether APOE $\varepsilon$4 genotype alone influences component expression, independent of demographic or diagnostic variables. Sessions were grouped as $\varepsilon$4 carriers (having at least one $\varepsilon$4 allele) or non-carriers, reflecting established genetic risk classification.

Among the 347 site-corrected sessions used in the earlier LMM analysis (Section II-E), APOE information was available for 346 sessions, including 145 $\varepsilon$4 carrier sessions (57 CN, 35 MCI, 53 AD) and 201 non-carrier sessions (110 CN, 61 MCI, 30 AD). Group differences in IC loadings were assessed using two-sample $t$-tests, conducted separately for each independent component that showed a significant diagnostic effect. A Bonferroni-corrected threshold ($\alpha = 0.0045$) was applied to account for multiple comparisons across the 11 components.

## III. Results

Joint ICA of the fused FBP-PET and FNC data identified eleven stable multimodal components, reflecting distinct patterns of amyloid deposition and corresponding functional connectivity across 552 sessions. Among these, nine PET-derived subcomponents represented distinct spatial patterns of amyloid accumulation across known brain regions or networks (Fig. 2). These components are available at https://github.com/Bikesh14/fMRI-PET-multimodal-fusion/tree/main/independent-components. Two components (IC1 and IC2) were localized to white matter (WM) regions. The remaining seven PET components showed strong correspondence with functional domains defined in the Neuromark 2.2 atlas [14], including: the cerebellar domain (CB; IC3), sensorimotor domain (SM; IC4, IC5), visual domain (VIS; IC6), frontal subdomain (FR; IC7), extended hippocampal subdomain (EH; IC8), and the default mode network subdomain (DMN; IC9).

*a) FNC components highlight both network disruption and compensatory behavior:* FNC patterns of the ICs revealed distinct disruptions and potential compensatory changes related to amyloid burden (Fig. 2). In IC4, increased amyloid accumulation in the sensorimotor (SM) domain was associated with relatively neutral intra-domain connectivity but stronger correlations with other domains and subdomains (Fig. 2(d)). In IC6, with increased amyloid deposition in visual regions, the component showed a pattern of anticorrelation among visual subdomains, while connections between visual and cerebellar regions appeared relatively stronger (Fig. 2(f)), suggesting potential compensatory cross-network interactions.

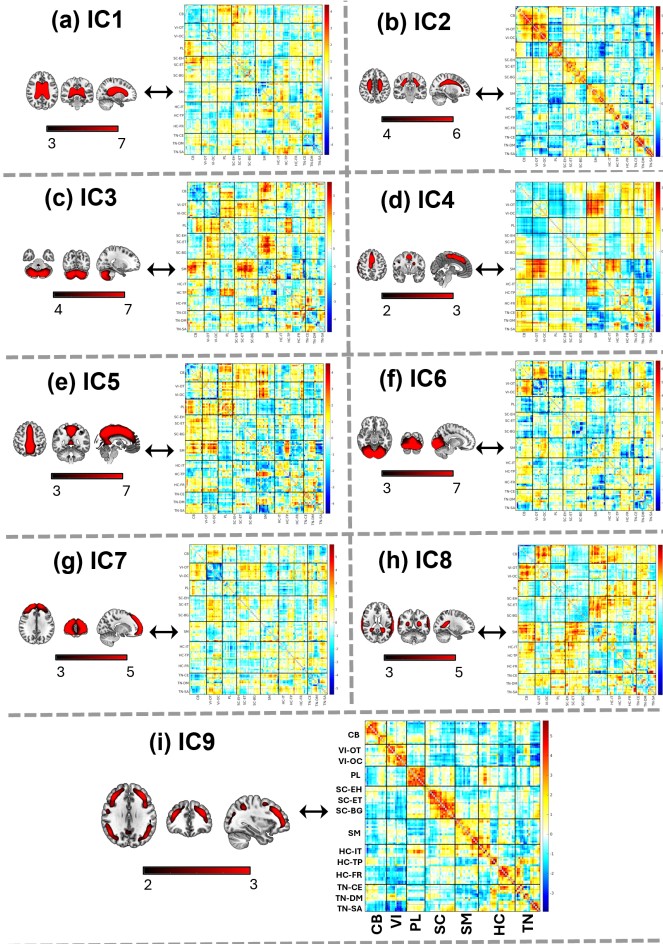

Fig. 2. Multimodal ICs showing PET (amyloid-β deposition) and corresponding FNC representations for Alzheimer's-related networks. PET maps correspond to: (a) white matter, (b) white matter, (c) cerebellar, (d) sensorimotor, (e) sensorimotor, (f) visual, (g) frontal, (h) extended hippocampal, and (i) default mode network regions.

IC7, linked to prefrontal amyloid deposition, showed disrupted connectivity within visual networks, indicating cross-network interference (Fig. 2(g)). In IC9, increased amyloid deposition in DMN regions, particularly in the AD group, was associated with stronger anticorrelation between the DMN and central executive (CE) network (Fig. 2(i)).

*b) Component loadings reveal diagnostic differences across white and grey matter regions:* LMM revealed significant group differences in IC session loadings (Fig. 3), with multiple comparisons corrected using Bonferroni adjustment across the 11 components (adjusted significance threshold: $\alpha = 0.0045$). Components highlighting white matter and cerebellar regions (IC1, IC2, IC3) showed decreasing loadings from CN to MCI to AD, while components localized to grey matter regions (IC5, IC7, IC8, IC9) showed increasing loadings with disease severity. The sensorimotor component (IC5), frontal component (IC7), and extended hippocampal component (IC8) demonstrated significant differences across all three group comparisons (MCI–CN, AD–CN, and AD–

MCI). To highlight, the DMN component (IC9) differed significantly in the AD–CN and AD–MCI comparisons, but not in the MCI–CN comparison ($p > \alpha$). We also observed that IC1, while showing an overall decreasing trend similar to IC2 and IC3, did not show a significant difference in the AD–MCI comparison ($p > \alpha$).

*c) Component loadings show significant associations with APOE ε4 carrier status:* When evaluated independently as a post hoc analysis, APOE ε4 carrier status also showed significant associations with the multimodal component loadings. All of the components that showed diagnostic group effects, except IC1, showed consistent patterns of significant differences in loadings between ε4 carriers and non-carriers, i.e., ε4 carriers showing similar directions of change in loadings (Fig. 4) as observed in the AD group from the diagnostic comparison (Fig. 3). These differences were also seen when analyzing only the sessions from individuals in the AD group. These findings indicate that: (1) overall ε4 carriers have a higher chance of experiencing disrupted brain connectivity and Aβ deposition as observed in the AD group; and (2) APOE ε4 also contributes to variation in component loadings within individual diagnostic groups, hinting at a possibility of genetic-based heterogeneity.

## IV. DISCUSSION

This study highlights the value of integrating PET and FNC to better understand the relationship between amyloid pathology and network-level dysfunction in AD. By applying joint ICA to multimodal data, we identified distinct patterns of amyloid deposition and functional connectivity that reflect co-occurring structural degeneration and adaptive neural mechanisms.

The progressive decline in white matter (IC1, IC2) and cerebellar (IC3) component weights across disease stages (CN > MCI > AD) likely reflects structural degeneration rather than amyloid-specific changes [16]. While FBP targets amyloid plaques, it also binds nonspecifically to myelin in white matter [17]. In the early stages, intact myelin leads to a higher signal, but as AD progresses, demyelination and axonal loss reduce tracer binding, resulting in a lower PET signal. The cerebellar region appears to exhibit similar characteristics to white matter, potentially undergoing myelin degeneration that contributes to reduced tracer uptake as the disease progresses. Together, these patterns suggest that even in regions not classically associated with early amyloid pathology, FBP-PET signal changes may offer indirect but meaningful markers of disease progression. The consistent decline in component loadings across stages (Fig. 3) further supports their potential utility as sensitive indicators of global neurodegenerative processes in AD. In contrast, remaining components highlighting gray matter regions—including sensorimotor (IC5), frontal (IC7), extended hippocampal (IC8), and DMN (IC9)—exhibited increasing amyloid deposition across disease stages, consistent with previous unimodal PET findings [3].

Our multimodal fusion approach revealed unique covarying patterns between regional amyloid deposition and network

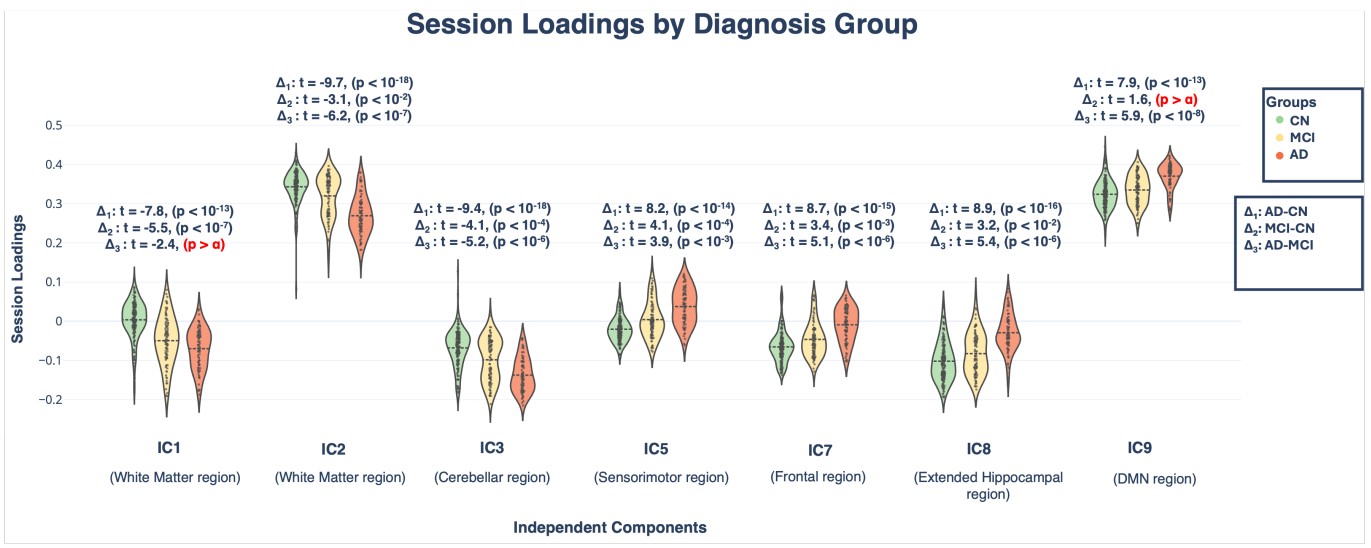

Fig. 3. Violin plots of session loadings across diagnostic groups (CN, MCI, AD) for multimodal ICs. Each IC is labeled by its dominant spatial region. Group differences ($\Delta_1$: AD–CN, $\Delta_2$: MCI–CN, $\Delta_3$: AD–MCI) were assessed using linear mixed-effects models. Statistical significance was evaluated using $\alpha = 0.0045$ (Bonferroni corrected threshold).

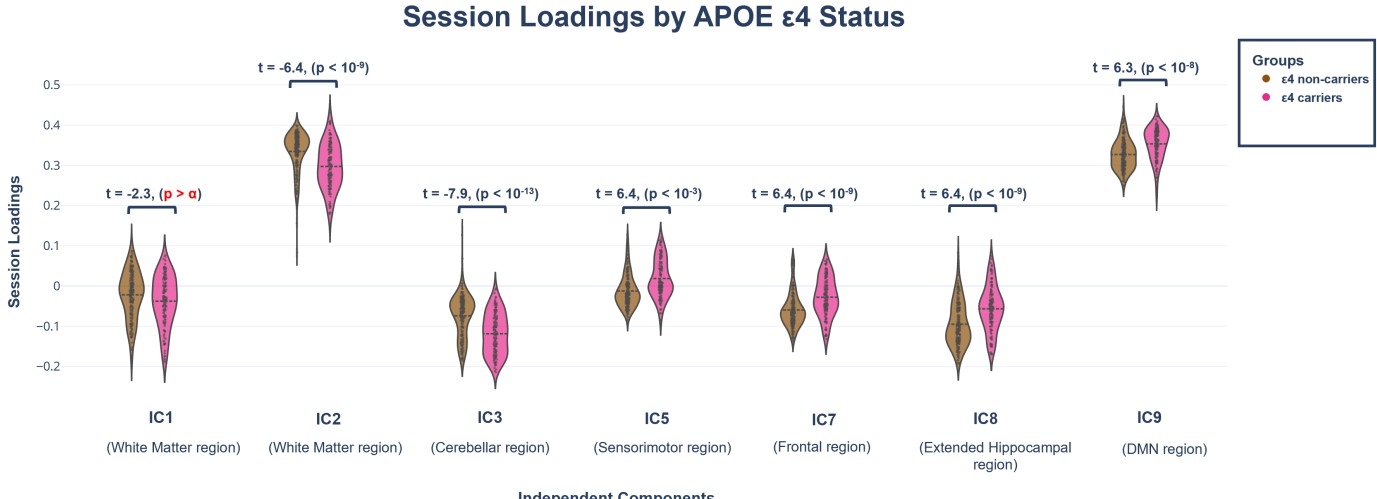

Fig. 4. Violin plots of session loadings by APOE $\varepsilon4$ status ($\varepsilon4$ carriers vs. non-carriers) for multimodal ICs. Group differences were assessed using two-sample $t$-tests. Statistical significance was evaluated using $\alpha = 0.0045$ (Bonferroni corrected threshold).

connectivity, offering insights not readily captured by uni-modal analyses. For example, increased A$\beta$ accumulation in the sensorimotor (SM) domain was associated with relatively neutral intra-domain connectivity but stronger correlations with other domains and subdomains (Fig. 2(d)), suggesting potential task rerouting or adaptive cross-network engagement to preserve SM function. Functional connectivity patterns further reflected a mix of adaptive and disruptive behaviors. The visual domain (IC6) showed stronger anticorrelations within the network but increased coupling with cerebellar regions, consistent with prior findings of enhanced FNC between visual and cerebellar domains in AD [4]. In the DMN-related component (IC9), increased amyloid-$\beta$ deposition was associated with a stronger anticorrelation between the DMN and the central ex-

ecutive subdomain. While such anticorrelation between these two subdomains is characteristic of healthy brain interactions, its persistence or amplification in the presence of elevated amyloid levels may reflect a compensatory mechanism, where the brain attempts to maintain functional segregation between task-positive and task-negative networks despite accumulating pathology.

Additionally, significant group differences in IC9 (DMN) were observed only between CN and AD, not between CN and MCI, supporting the notion that DMN disruption manifests in later disease stages. This aligns with previous work [18], which suggested DMN as one of the last networks to be affected in AD. In contrast, IC1 (associated with white matter) showed a significant difference between MCI and CN, but not

between AD and MCI. This suggests that the major changes in this component may happen early in the disease, possibly during the transition from normal aging to MCI, with no significant changes in later stages. These findings highlight the potential of understudied WM for the early detection of AD-related changes.

The observed differences in component loadings between APOE $\varepsilon$4 carriers and non-carriers suggest that APOE genotype may influence how these multimodal brain patterns are expressed. These differences were not solely tied to clinical diagnosis, indicating that genetic factors like APOE status may play a role in shaping individual variability in the combined expression of amyloid accumulation and functional network changes. Incorporating such genetic information may improve the interpretation of multimodal brain signatures and promise further strengthened imaging-genomic biomarkers for stratification in Alzheimer's disease studies.

By fusing PET and FNC, we captured shared variations between amyloid burden and network dysfunction that are often missed by unimodal approaches. In future work, we plan to validate these findings in larger longitudinal cohorts and extend the framework to incorporate additional modalities such as tau-PET, CSF biomarkers, and genomics. Overall, this integrative framework offers a promising direction for improving early detection, enhancing patient stratification, and deepening our understanding of Alzheimer's disease progression.

## V. Conclusion

Our study highlights the potential of multimodal fusion to improve early detection and characterization of Alzheimer's disease. Through joint independent component analysis of combined amyloid-PET and functional network connectivity data, we identified integrated biomarkers that reveal the interplay between amyloid pathology and functional disruption, offering insights not accessible through unimodal approaches. This framework offers a more comprehensive view of disease progression and supports the development of more sensitive diagnostic tools.

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
