# OpenReview forum: "Multimodal Fusion Analysis of [18F]Florbetapir PET  and Multiscale Functional Network Connectivity in  Alzheimer’s Disease"
_IEEE.org/EMBS/BHI/2025/Conference — BHI 2025_

### Official Review · Reviewer_Sdj1 · 2025-07-03
**A neat study**

**Confidence:** 4
**Clarity Of Writing:** great
**Clinical Significance:** great
**Methodological Novelty:** great
**Overall Rating:** 5
**Final Rating:** 6

**Experiments And Results:**

great

**Questions For The Authors:**

Some concerns:
- Matching of PET and fMRI scans are done by using a 180d window. While there might not be a clear accepted standard yet for multimodal studies involving only functional modalities (as opposed to CT/MRI studies which typically use 1 year), could this be an issue since significant functional changes might occur within a smaller time window?
- PET features seem at least 10 times larger than FNC features, despite downsampling the PET data. While it might not necessarily be a major issue in this context (based on the methodology adopted), having additional experiments that demonstrate that this imbalance does not adversely affect the biomarkers reported could ally any concerns.
- Table 1, should it be ‘dementia’ or “AD”? AD is a particular subtype of dementia and it is more likely the case that ADNI is focused on AD.
- While not a factor that influences the score, do consider publicly sharing precise details about the ICs (e.g. nii files) so that other researchers could use them in future e.g. for meta-analyses.
- Having a replication dataset that further supports the current findings would make them much stronger (especially for a new cohort with different demographics/racial composition), but it is understandable if the authors find that to be too much (not easy due to heterogeneity).

**Strengths:**

- Overall, a neat study that takes care of typical pitfalls and no major concerns about methodology was found.
- Choice of PET and FC is relatively under-explored and the analysis has produced several interesting findings.

**Summary Of The Paper:**

A joint ICA approach of studying potential PET and FC biomarkers of AD is presented. Experiments are conducted on the ADNI dataset, including fitting a LME model and a group-level comparison of APOE e4 status.

**Weaknesses:**

- Limited methodological novelty - more of an application of existing tools on novel multimodal combinations.
- Multimodal fusion is limited to concatenation of features and there are some small concerns (starkly different lengths, time gap between scans).
- While interesting, findings were only demonstrated on 1 dataset and generalisability is uncertain.

---

### Official Review · Reviewer_6qcP · 2025-07-11
**fusion analysis allows for linking deposition of amyloid beta plaques (PET) to connectivity within the brain (rsfMRI)**

**Confidence:** 3
**Clarity Of Writing:** great
**Clinical Significance:** good
**Methodological Novelty:** good
**Overall Rating:** 7
**Final Rating:** 7

**Experiments And Results:**

great

**Questions For The Authors:**

minor corrections could help this paper flow a bit better. these changes does not affect my score for this paper.

1. Flow in Results (III). maybe IIIa and IIIc should be together as both of them are talking about session loadings followed by IIIb (or b,a,c order). Also Fig 2 should follow appropriately however you decide (session loading figures: fig 3 and fig 4)

2. Fig 2 could be made wider and maybe span two columns. i had a hard time squinting the legends of the FNC representation. Should be doable without going over the page limit (of 7 pages?)

3. Fig 3 violin plots have 10^-X and 0.002 (in IC2 and IC8), might be better to standardize all to a 10^ format

4. Fig 3 legend notation for Delta-1, -2, 3 is Dementia-CN, MCI-CN, and Dementia-MCI, whereas in the text of IIIa) its flipped: CN-MCI, CN-Dementia. It may be best to standardize them to be the same

5. all references have the doi except for [2]

**Strengths:**

There is sufficient motivation for the reader to understand why this research was needed. The paper is well written and explains in detail the research methodology. It explains in detail how the joint ICA is performed, how the main ICA components are identified and proposes the session loading and how session loading is used to tie with the diagnosis and the APOE genotype. The results for session loadings for both the diagnosis and the APOE genotype clearly shows that there are significant changes in the CN-MCI-dementia groups and between e4 carriers and non carriers.

**Summary Of The Paper:**

The paper proposes fusing information between two modalities (PET and rsfMRI) using joint ICA where the PET data is enhanced with amyloid specific radiotracers like FBP to detect amyloid beta plaques that is indicative of Alzheimers. On the other hand, rsfMRI data is used to derive the Functional Network Connectivity of the brain which also degrades with AD progression. By linking longitudinal PET data with fsMRI data the goal of the paper is to  identify differences in disease progression  (MCI and Dementia) and  in the congitively normal subjects and by linking the two modalities.

**Weaknesses:**

i do not find any major weakness in the research methodology or the motivation.

a minor thing that could improve this paper would be to expand IIIb) where there is a discussion of the FNC components and compensatory behavior to also include parts from the discussion (para 3) and list all places where disruptions are occurring and compensatory behavior is being observed

---

### Official Review · Reviewer_e3vA · 2025-07-13
**Integration of resting state fMRI and FBP-PET analyzed with Independent Component Analysis and with some genetic correlation**

**Confidence:** 3
**Clarity Of Writing:** good
**Clinical Significance:** good
**Methodological Novelty:** good
**Overall Rating:** 7
**Final Rating:** 7

**Experiments And Results:**

good

**Questions For The Authors:**

It was mentioned in the beginning of the paper, but would this work (or really, future work) benefit from using a similar approach for task-based fMRI integrated with FBP-PET?

**Strengths:**

The novelty of this work is the independent component analysis of PET and fMRI scans fairly close temporally of a fairly large amount of participants.

**Summary Of The Paper:**

In this paper, the authors integrate PET-based amyloid plaque images with functional MRI images taken at fairly close timepoints and analyze for patterns of plaque deposition together with alterations in resting-state function using joint independent component analysis. This reveals patterns of disruption as well as patterns of compensatory connections made afterwards, as well as patterns of degeneration with time along with APOE ε4 carrier status.

**Weaknesses:**

I agree with the authors in that further work would benefit from both more scans and scans that are more closely associated temporally. 180 days used as the limit between scans could hopefully be reduced and help show greater association between the plaque deposits seen with the functional changes observed. The genetic component is a nice inclusion but would probably benefit from further analyses and markers, as the authors mention.

---

### Official Review · Reviewer_q3Dh · 2025-07-20
**Review of Submission141**

**Confidence:** 4
**Clarity Of Writing:** great
**Clinical Significance:** great
**Methodological Novelty:** good
**Overall Rating:** 7

**Experiments And Results:**

great

**Questions For The Authors:**

What is the benefit of multimodal fusion over unimodal analysis in this study? Did joint ICA produce components that were not detectable using PET or fMRI alone?

Were any components truly cross-modal? Can you show examples where both PET and FNC contributed meaningfully to the same component?

How did you handle feature imbalance between PET voxels and FNC edges? Was any modality dominating the ICA decomposition despite z-scoring?

Did you assess predictive value of the components? Any classification or ROC analyses to show how well component loadings separate CN, MCI, and Dementia?

**Strengths:**

This study presents several notable strengths:
(1) it employs a multimodal fusion framework using joint ICA, enabling integrated analysis of PET and fMRI data at the feature level rather than relying on post hoc associations;
(2) it utilizes a large, longitudinal dataset (552 matched sessions from 395 subjects), enhancing the generalizability of the findings;
(3) the use of ICASSO stability analysis ensures robust and reliable component extraction;
(4) it identifies distinct biomarkers associated with early and late stages of Alzheimer’s disease, including novel early markers in white matter;
(5) the analysis incorporates genetic stratification via APOE ε4 status, adding a valuable genomic dimension to the biomarker interpretation;
(6) the study offers biologically meaningful interpretations of PET and FNC patterns, highlighting both compensatory and degenerative network behaviors relevant to disease progression.

**Summary Of The Paper:**

This study presents a multimodal fusion analysis combining [18F]Florbetapir PET imaging and resting-state fMRI-derived functional network connectivity (FNC) to investigate Alzheimer’s disease (AD). Using joint Independent Component Analysis (jICA) on 552 matched PET-fMRI sessions from cognitively normal, MCI, and dementia patients, the authors identified 11 independent components reflecting co-occurring patterns of amyloid-beta deposition and functional connectivity changes. Results showed that white matter and cerebellar components declined with disease progression, indicating early structural degeneration, while gray matter components—particularly in sensorimotor, frontal, hippocampal, and default mode network regions—showed increased amyloid accumulation and altered connectivity associated with later stages. Significant differences in component loadings were also observed between APOE ε4 carriers and non-carriers, suggesting genotype-linked heterogeneity in disease expression. Overall, this integrative approach uncovered distinct multimodal biomarkers that advance understanding of AD progression beyond what unimodal methods can offer

**Weaknesses:**

A central weakness of the paper lies in its lack of empirical benchmarking to justify the use of multimodal fusion over unimodal approaches. While the authors posit that joint ICA captures covarying patterns between amyloid deposition and functional connectivity, they do not demonstrate whether this integration yields superior or complementary insights compared to analyzing PET or fMRI data separately. For example, early-stage biomarkers like IC1 and IC2 appear to be primarily PET-driven, yet no comparative analysis is provided to assess whether these components could have been equivalently or more clearly identified using PET data alone. Similarly, while functional connectivity alterations are discussed in later components, no quantitative evidence is presented to show that including PET data improved the sensitivity or interpretability of these findings. Without a baseline from unimodal ICA or other fusion techniques (e.g., CCA, simple concatenation), it remains unclear whether joint ICA offers a meaningful advantage beyond its conceptual appeal. A thorough benchmarking experiment — comparing unimodal and multimodal results on metrics such as diagnostic group separability, stability, and biological interpretability — would help validate the claimed benefits of the fusion approach and ensure that the added complexity is scientifically justified.